# DEFORMABLE KERNELS: ADAPTING EFFECTIVE RECEPTIVE FIELDS FOR OBJECT DEFORMATION

**Hang Gao**[1,3*], **Xizhou Zhu**[2,3*], **Steve Lin**[3], **Jifeng Dai**[3]
[1]UC Berkeley [2]University of Science and Technology of China [3]Microsoft Research Asia
`hangg@eecs.berkeley.edu, ezra0408@mail.ustc.edu.cn`
`{stevelin,jifdai}@microsoft.com`
[http://people.eecs.berkeley.edu/~hangg/deformable-kernels/](http://people.eecs.berkeley.edu/~hangg/deformable-kernels/)

## ABSTRACT

Convolutional networks are not aware of an object's geometric variations, which leads to inefficient utilization of model and data capacity. To overcome this issue, recent works on deformation modeling seek to spatially reconfigure the data towards a common arrangement such that semantic recognition suffers less from deformation. This is typically done by augmenting static operators with learned free-form sampling grids in the image space, dynamically tuned to the data and task for adapting the receptive field. Yet adapting the receptive field does not quite reach the actual goal – what really matters to the network is the *effective* receptive field (ERF), which reflects how much each pixel contributes. It is thus natural to design other approaches to adapt the ERF directly during runtime.

In this work, we instantiate one possible solution as Deformable Kernels (DKs), a family of novel and generic convolutional operators for handling object deformations by directly adapting the ERF while leaving the receptive field untouched. At the heart of our method is the ability to resample the original kernel space towards recovering the deformation of objects. This approach is justified with theoretical insights that the ERF is strictly determined by data sampling locations and kernel values. We implement DKs as generic drop-in replacements of rigid kernels and conduct a series of empirical studies whose results conform with our theories. Over several tasks and standard base models, our approach compares favorably against prior works that adapt during runtime. In addition, further experiments suggest a working mechanism orthogonal and complementary to previous works.

## 1 INTRODUCTION

The rich diversity of object appearance in images arises from variations in object semantics and deformation. Semantics describe the high-level abstraction of what we perceive, and deformation defines the geometric transformation tied to specific data (Gibson, 1950). Humans are remarkably adept at making abstractions of the world (Hudson & Manning, 2019); we see in raw visual signals, abstract semantics away from deformation, and form concepts.

Interestingly, modern convolutional networks follow an analogous process by making abstractions through local connectivity and weight sharing (Zhang, 2019). However, such a mechanism is an inefficient one, as the emergent representations encode semantics and deformation together, instead of as disjoint notions. Though a convolution responds accordingly to each input, how it responds is primarily programmed by its rigid kernels, as in Figure 1(a, b). In effect, this consumes large model capacity and data modes (Shelhamer et al., 2019).

We argue that the awareness of deformations emerges from adaptivity – the ability to adapt at runtime (Kanazawa et al., 2016; Jia et al., 2016; Li et al., 2019). Modeling of geometric transformations has been a constant pursuit for vision researchers over decades (Lowe et al., 1999; Lazebnik et al., 2006; Jaderberg et al., 2015; Dai et al., 2017). A basic idea is to spatially recompose data towards a common mode such that semantic recognition suffers less from deformation. A recent work that

---

*Equal contributions. Work is done when Hang and Xizhou are interns at Microsoft Research Asia.

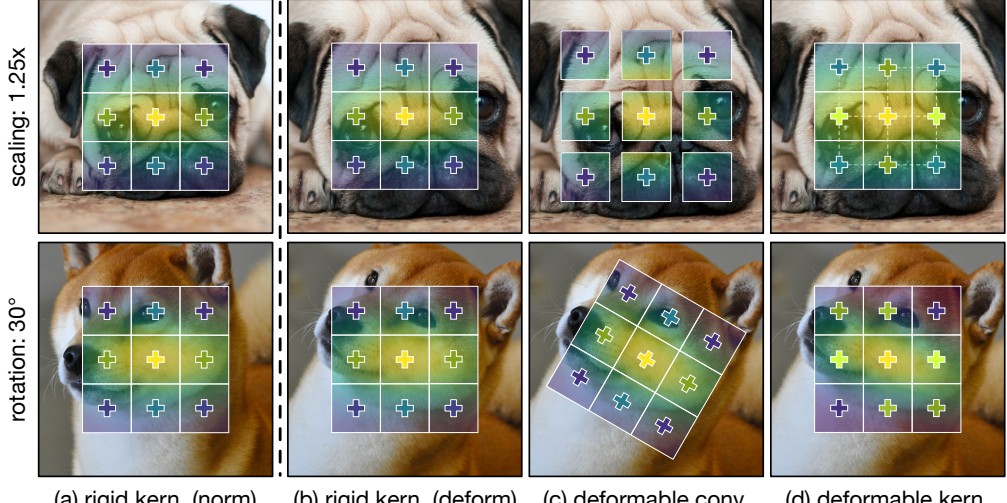

Figure 1: **Adaptation for deformation.** We show how different $3 \times 3$ convolutions interact with deformations of two images. Kernel spaces are visualized as flat 2D Gaussians. Each "+" indicates a computation between a pixel and a kernel value sampled from the data and kernel space. Their colors represent corresponding kernel values. **(a, b)** Rigid kernels cannot adapt to specific deformations, thus consuming large model and data capacity. **(c)** Deformable Convolutions (Dai et al., 2017) reconfigure data towards common arrangement to counter the effects of geometric deformation. **(d)** Our Deformable Kernels (DKs) instead resample kernels and, in effect, adapt kernel spaces while leaving the data untouched. Note that (b) and (c) share kernel values but sample different data locations, while (b) and (d) share data locations but sample different kernel values.

is representative of this direction is Deformable Convolution (Dai et al., 2017; Zhu et al., 2019). As shown in Figure 1(c), it augments the convolutions with free-form sampling grids in the data space. It is previously justified as adapting receptive field, or what we phrase as the "theoretical receptive field", that defines *which* input pixels can contribute to the final output. However, theoretical receptive field does not measure *how much* impact an input pixel actually has. On the other hand, Luo et al. (2016) propose to measure the effective receptive field (ERF), i.e. the partial derivative of the output with respect to the input data, to quantify the exact contribution of each raw pixel to the convolution. Since adapting the theoretical receptive field is not the goal but a means to adapt the ERF, why not directly tune the ERF to specific data and tasks at runtime?

Toward this end, we introduce Deformable Kernels (DKs), a family of novel and generic convolutional operators for deformation modeling. We aim to augment rigid kernels with the expressiveness to directly interact with the ERF of the computation during inference. Illustrated in Figure 1(d), DKs learn free-form offsets on kernel coordinates to deform the original kernel space towards specific data modality, rather than recomposing data. This can directly adapt ERF while leaving receptive field untouched. The design of DKs that is agnostic to data coordinates naturally leads to two variants – the global DK and the local DK, which behave differently in practice as we later investigate. We justify our approach with theoretical results which show that ERF is strictly determined by data sampling locations and kernel values. Used as a generic drop-in replacement of rigid kernels, DKs achieve empirical results coherent with our developed theory. Concretely, we evaluate our operator with standard base models on image classification and object detection. DKs perform favorably against prior works that adapt during runtime. With both quantitative and qualitative analysis, we further show that DKs can work orthogonally and complementarily with previous techniques.

## 2 RELATED WORKS

We distinguish our work within the context of deformation modeling as our goal, and dynamic inference as our means.

**Deformation Modeling:** We refer to deformation modeling as learning geometric transformations in 2D image space without regard to 3D. One angle to attack deformation modeling is to craft certain geometric invariances into networks. However, this usually requires designs specific to certain kinds of deformation, such as shift, rotation, reflection and scaling (Sifre & Mallat, 2013; Bruna & Mallat, 2013; Kanazawa et al., 2016; Cohen & Welling, 2016; Worrall et al., 2017; Esteves et al., 2018). Another line of work on this topic learns to recompose data by either semi-parameterized or completely free-form sampling in image space: Spatial Transformers (Jaderberg et al., 2015) learns 2D affine transformations, Deep Geometric Matchers (Rocco et al., 2017) learns thin-plate spline transformations, Deformable Convolutions (Dai et al., 2017; Zhu et al., 2019) learns free-form transformations.

We interpret sampling data space as an effective approach to adapt effective receptive fields (ERF) by directly changing receptive field. At a high-level, our Deformable Kernels (DKs) share intuitions with this line of works for learning geometric transformations, yet are instantiated by learning to sample in kernel space which directly adapt ERF while leaving theoretical receptive fields untouched. While kernel space sampling is also studied in Deformable Filter (Xiong et al., 2019) and KPConv (Thomas et al., 2019), but in their contexts, sampling grids are computed from input point clouds rather than learned from data corpora.

**Dynamic Inference:** Dynamic inference adapts the model or individual operators to the observed data. The computation of our approach differs from self-attention (Vaswani et al., 2017; Wang et al., 2018) in which linear or convolution modules are augmented with subsequent queries that extract from the same input. We consider our closest related works in terms of implementation as those approaches that adapt convolutional kernels at run time. It includes but is not limited to Dynamic Filters (Jia et al., 2016), Selective Kernels (Li et al., 2019) and Conditional Convolutions (Yang et al., 2019). All of these approaches can learn and infer customized kernel spaces with respect to the data, but are either less inefficient or are loosely formulated. Dynamic Filters generate new filters from scratch, while Conditional Convolutions extend this idea to linear combinations of a set of synthesized filters. Selective Kernels are, on the other hand, comparably lightweight, but aggregating activations from kernels of different size is not as compact as directly sampling the original kernel space. Another line of works contemporary to ours (Shelhamer et al., 2019; Wang et al., 2019) is to compose free-form filters with structured Gaussian filters, which essentially transforms kernel spaces by data. Our DKs also differ from these works with the emphasize of direct adaptation the ERF rather than the theoretical receptive field. As mentioned previously, the true goal should be to adapt the ERF, and to our knowledge, our work is the first to study dynamic inference of ERFs.

## 3 APPROACH

We start by covering preliminaries on convolutions, including the definition of effective receptive field (ERF). We then formulate a theoretical framework for analyzing ERFs, and thus motivate our Deformable Kernels (DKs). We finally elaborate different DK variants within such a framework. Our analysis suggests compatibility between DKs and the prior work.

### 3.1 A DIVE INTO CONVOLUTIONS

**2D Convolution:** Let us first consider an input image $\boldsymbol{I} \in \mathbb{R}^{D \times D}$. By convolving it with a kernel $\boldsymbol{W} \in \mathbb{R}^{K \times K}$ of stride 1, we have an output image $\boldsymbol{O}$ whose pixel values at each coordinate $\boldsymbol{j} \in \mathbb{R}^2$ can be expressed as

$$\boldsymbol{O_j} = \sum_{\boldsymbol{k} \in \mathcal{K}} \boldsymbol{I_{j+k}} \boldsymbol{W_k}, \tag{1}$$

by enumerating discrete kernel positions $\boldsymbol{k}$ within the support $\mathcal{K} = [-K/2, K/2]^2 \cap \mathbb{Z}$. This defines a rigid grid for sampling data and kernels.

**Theoretical Receptive Field:** The same kernel $\boldsymbol{W}$ can be stacked repeatedly to form a *linear* convolutional network with $n$ layers. The theoretical receptive field can then be imagined as the "accumulative coverage" of kernels at each given output unit on the input image by deconvolving back through the network. This property characterizes a set of input fields that could fire percepts onto corresponding output pixels. The size of a theoretical receptive field scales linearly with respect to the network depth $n$ and kernel size $K$ (He et al., 2016).

**Effective Receptive Field:** Intuitively, not all pixels within a theoretical receptive field contribute equally. The influence of different fields varies from region to region thanks to the central emphasis of stacked convolutions and also to the non-linearity induced by activations. The notion of *effective receptive field* (ERF) (Luo et al., 2016) is thus introduced to measure the impact of each input pixel on the output at given locations. It is defined as a partial derivative field of the output with respect to the input data. With the numerical approximations in linear convolution networks, the ERF was previously identified as a Gaussian-like soft attention map over input images whose size grows *fractionally* with respect to the network depth $n$ and linearly to the kernel size $K$. Empirical results validate this idea under more complex and realistic cases when networks exploit non-linearities, striding, padding, skip connections, and subsampling.

## 3.2 ANALYSIS ON EFFECTIVE RECEPTIVE FIELDS

We aim to revisit and complement the previous analysis on ERFs by Luo et al. (2016). While the previous analysis concentrates on studying the expectation of an ERF, i.e., when network depth $n$ approaches infinity or all kernels are randomly distributed without learning in general, our analysis focuses on how we can perturb the computation such that the change in ERF is predictable, given an input and a set of kernel spaces.

We start our analysis by considering a *linear* convolutional network, without any unit activations, as defined in Section 3.1. For consistency, superscripts are introduced to image $\boldsymbol{I}$, kernel $\boldsymbol{W}$, and subscripts to kernel positions $\boldsymbol{k}$ to denote the index $s \in [1, n]$ of each layer. Formally, given an input image $\boldsymbol{I}^{(0)}$ and a set of $K \times K$ kernels $\{\boldsymbol{W}^{(s)}\}_{s=1}^n$ of stride 1, we can roll out the final output $\boldsymbol{O} \equiv \boldsymbol{I}^{(n)}$ by unfolding Equation 1 as

$$
\boldsymbol{I}_{\boldsymbol{j}}^{(n)} = \sum_{\boldsymbol{k}_n \in \mathcal{K}} \boldsymbol{I}_{\boldsymbol{j}+\boldsymbol{k}_n}^{(n-1)} \boldsymbol{W}_{\boldsymbol{k}_n}^{(n)} = \sum_{(\boldsymbol{k}_{n-1}, \boldsymbol{k}_n) \in \mathcal{K}^2} \boldsymbol{I}_{\boldsymbol{j}+\boldsymbol{k}_n+\boldsymbol{k}_{n-1}}^{(n-2)} \boldsymbol{W}_{\boldsymbol{k}_n}^{(n)} \boldsymbol{W}_{\boldsymbol{k}_{n-1}}^{(n-1)} = \cdots
$$
$$
= \sum_{(\boldsymbol{k}_1, \boldsymbol{k}_2, \ldots, \boldsymbol{k}_n) \in \mathcal{K}^n} \left( \boldsymbol{I}_{\boldsymbol{j}+\sum_{s=1}^n \boldsymbol{k}_s}^{(0)} \cdot \prod_{s=1}^n \boldsymbol{W}_{\boldsymbol{k}_s}^{(s)} \right). \tag{2}
$$

By definition[1], the effective receptive field value $\mathcal{R}^{(n)}(\boldsymbol{i}; \boldsymbol{j}) \equiv \partial \boldsymbol{I}_{\boldsymbol{j}}^{(n)} / \partial \boldsymbol{I}_{\boldsymbol{i}}^{(0)}$ of output coordinate $\boldsymbol{j}$ that takes input coordinate $\boldsymbol{i}$ can be computed by

$$
\mathcal{R}^{(n)}(\boldsymbol{i}; \boldsymbol{j}) = \sum_{(\boldsymbol{k}_1, \boldsymbol{k}_2, \ldots, \boldsymbol{k}_n) \in \mathcal{K}^n} \left( \mathbb{1}\left[\boldsymbol{j} + \sum_{s=1}^n \boldsymbol{k}_s = \boldsymbol{i}\right] \cdot \prod_{s=1}^n \boldsymbol{W}_{\boldsymbol{k}_s}^{(s)} \right), \tag{3}
$$

where $\mathbb{1}[\cdot]$ denotes the indicator function. This result indicates that ERF is related only to the data sampling location $\boldsymbol{j}$, kernel sampling location $\boldsymbol{k}$, and kernel matrices $\{\boldsymbol{W}^{(s)}\}$.

If we replace the $m^{\text{th}}$ kernel $\boldsymbol{W}^{(m)}$ with a $1 \times 1$ kernel of a single parameter $\boldsymbol{W}_{\tilde{\boldsymbol{k}}_m}^{(m)}$ sampled from it, the value of ERF becomes to

$$
\mathcal{R}^{(n)}(\boldsymbol{i}; \boldsymbol{j}, \tilde{\boldsymbol{k}}_m) = \sum_{(\boldsymbol{k}_1, \ldots, \boldsymbol{k}_{m-1}, \boldsymbol{k}_{m+1}, \ldots, \boldsymbol{k}_n) \in \mathcal{K}^{n-1}} \left( \mathbb{1}\left[\boldsymbol{j} + \sum_{s \in \mathbb{S}} \boldsymbol{k}_s = \boldsymbol{i}\right] \cdot \prod_{s \in \mathbb{S}} \boldsymbol{W}_{\boldsymbol{k}_s}^{(s)} \cdot \boldsymbol{W}_{\tilde{\boldsymbol{k}}_m}^{(m)} \right), \tag{4}
$$

where $\mathbb{S} = [1, n] \setminus \{m\}$. Since a $K \times K$ kernel can be deemed as a composition of $K^2$ $1 \times 1$ kernels distributed on a square grid, Equation 3 can thus be reformulated as

$$
\mathcal{R}^{(n)}(\boldsymbol{i}; \boldsymbol{j}) = \sum_{\boldsymbol{k}_m \in \mathcal{K}} \mathcal{R}^{(n)}(\boldsymbol{i}; \boldsymbol{j} + \boldsymbol{k}_m, \boldsymbol{k}_m). \tag{5}
$$

For the case of complex non-linearities, where we here consider post ReLU[2] activations in Equation 1,

$$
\boldsymbol{O}_{\boldsymbol{j}} = \max(\sum_{\boldsymbol{k} \in \mathcal{K}} \boldsymbol{I}_{\boldsymbol{j}+\boldsymbol{k}} \boldsymbol{W}_{\boldsymbol{k}}, 0). \tag{6}
$$

---

[1] The original definition of ERF in Luo et al. (2016) focuses on the central coordinate of the output, i.e. $\boldsymbol{j} = (0, 0)$, to partially avoid the effects of zero padding. In this work, we will keep $\boldsymbol{j}$ in favor of generality while explicitly assuming input size $D \to \infty$.

[2] Our analysis currently only considers the ReLU network for its nice properties and prevalent popularity.

We can follow a similar analysis and derive corresponding ERF as

$$\mathcal{R}^{'(n)}(\boldsymbol{i};\boldsymbol{j},\tilde{\boldsymbol{k}}_m) = \sum_{(\boldsymbol{k}_1,\cdots,\boldsymbol{k}_{m-1},\boldsymbol{k}_{m+1},\cdots,\boldsymbol{k}_n)\in\mathcal{K}^{n-1}} \left( \mathcal{C}^{(n)}(\boldsymbol{i};\boldsymbol{j},\boldsymbol{k}_1,\cdots,\boldsymbol{k}_n,\tilde{\boldsymbol{k}}_m) \cdot \prod_{s\in\mathbb{S}} \boldsymbol{W}_{\boldsymbol{k}_s}^{(s)} \cdot \boldsymbol{W}_{\tilde{\boldsymbol{k}}_m}^m \right)$$

where $\mathcal{C}^{(n)}(\boldsymbol{i};\boldsymbol{j},\boldsymbol{k}_1,\cdots,\boldsymbol{k}_n,\tilde{\boldsymbol{k}}_m) = \mathbb{1}\big[\boldsymbol{j}+\sum_{s\in\mathbb{S}}\boldsymbol{k}_s=\boldsymbol{i}\big] \prod_{s\in\mathbb{S}} \mathbb{1}\big[\boldsymbol{I}_{\boldsymbol{j}}^{(s-1)}\boldsymbol{W}_{\boldsymbol{k}_s}^{(s)}>0\big]\mathbb{1}\big[\boldsymbol{I}_{\boldsymbol{j}}^{(m-1)}\boldsymbol{W}_{\boldsymbol{k}_m}^{(m)}>0\big].$

Here we can see that the ERF becomes data-dependent due to the coefficient $\mathcal{C}$, which is tied to input coordinates, kernel sampling locations, and input data $\boldsymbol{I}^{(0)}$. The more detailed analysis of this coefficient is beyond the scope of this paper. However, it should be noted that this coefficient only "gates" the contribution of the input pixels to the output. So in practice, ERF is "porous" – there are inactive (or gated) pixel units irregularly distributed around the ones that fire. This phenomenon also appeared in previous studies (such as in Luo et al. (2016), Figure 1). The maximal size of an ERF is still controlled by the data sampling location and kernel values as in the linear cases in Equation 5.

A nice property of Equation 4 and Equation 5 is that all computations are linear, making it compatible with any linear sampling operators for querying kernel values of fractional coordinates. In other words, sampling kernels in effect samples the ERF on the data in the linear case, but also roughly generalizes to non-linear cases as well. This finding motivates our design of Deformable Kernels (DKs) in Section 3.3.

### 3.3 DEFORMABLE KERNELS

In the context of Equation 1, we resample the kernel $\boldsymbol{W}$ with a group of *learned* kernel offsets denoted as $\{\Delta\boldsymbol{k}\}$ that correspond to each discrete kernel position $\boldsymbol{k}$. This defines our DK as

$$\boldsymbol{O}_{\boldsymbol{j}} = \sum_{\boldsymbol{k}\in\mathcal{K}} \boldsymbol{I}_{\boldsymbol{j}+\boldsymbol{k}}\boldsymbol{W}_{\boldsymbol{k}+\Delta\boldsymbol{k}}, \tag{7}$$

and the value of ERF as

$$\mathcal{R}_{\text{DK}}^{(n)}(\boldsymbol{i};\boldsymbol{j}) = \sum_{\boldsymbol{k}_m\in\mathcal{K}} \mathcal{R}^{(n)}(\boldsymbol{i};\boldsymbol{j}+\boldsymbol{k}_m,\boldsymbol{k}_m+\Delta\boldsymbol{k}_m). \tag{8}$$

Note that this operation leads to sub-pixel sampling in the kernel space. In practice, we use bilinear sampling to interpolate within the discrete kernel grid.

Intuitively, the size (resolution) of the original kernel space can affect sampling performance. Concretely, suppose we want to sample a $3 \times 3$ kernel. DKs do not have any constraint on the size of the original kernel space, which we call the "scope size" of DKs. That said, we can use a $\boldsymbol{W}$ of any size $K'$ even though the number of sampling locations is fixed as $K^2$. We can thus exploit large kernels – the largest ones can reach $9 \times 9$ in our experiments with nearly no overhead in computation since bilinear interpolations are extremely lightweight compared to the cost of convolutions. This can also increase the number of learning parameters, which in practice might become intractable if not handled properly. In our implementation, we will exploit depthwise convolutions (Howard et al., 2017) such that increasing scope size induces a negligible amount of extra parameters.

As previously discussed, sampling the kernel space in effect transforms into sampling the ERF. On the design of locality and spatial granularity of our learned offsets, DK naturally delivers two variants – the global DK and the local DKs, as illustrated in Figure 2. In both operators, we learn a kernel offset generator $\mathcal{G}$ that maps an input patch into a set of kernel offsets that are later applied to rigid kernels.

In practice, we implement $\mathcal{G}_{\text{global}}$ as a stack of one global average pooling layer, which reduces feature maps into a vector, and another fully-connected layer without non-linearities, which projects the reduced vector into an offset vector of $2K^2$ dimensions. Then, we apply these offsets to *all* convolutions for the input image following Equation 7. For local DKs, we implement $\mathcal{G}_{\text{local}}$ as an extra convolution that has the same configuration as the target kernel, except that it only has $2K^2$ output channels. This produces kernel sampling offsets $\{\Delta\boldsymbol{k}\}$ that are additionally indexed by output locations $\boldsymbol{j}$. It should be noted that similar designs were also discussed in Jia et al. (2016), in which filters are generated given either an image or individual patches from scratch rather than by resampling.

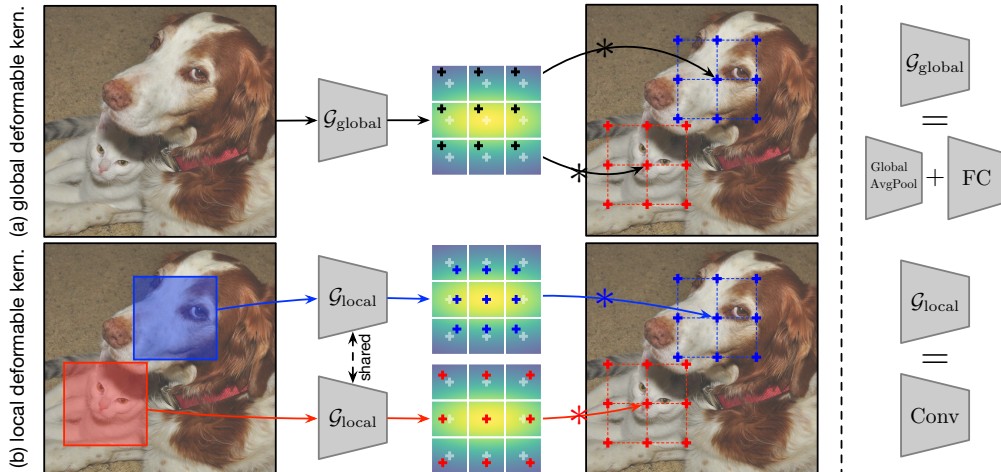

Figure 2: **Instantiations.** We show how DK variants works with an example image that contains a large and a small object. **(a)** The global DK learns one set of kernel sampling grid given an input image and apply it to all data positions. **(b)** The local DK adapts kernels for each input patches, and induces better locality for deformation modeling.

Intuitively, we expect the global DK to adapt kernel space between different images but not within a single input. The local DK can further adpat to specific image patches: for smaller objects, it is better to have shaper kernels and thus denser ERF; for larger objects, flatter kernels can be more beneficial for accumulating a wider ERF. On a high level, local DKs can preserve better locality and have larger freedom to adapt kernel spaces comparing to its global counterpart. We later compare these operators in our experiments.

### 3.4 LINK WITH DEFORMABLE CONVOLUTIONS

The core idea of DKs is to learn adaptive offsets to sample the kernel space for modeling deformation, which makes them similar to Deformable Convolutions (Dai et al., 2017; Zhu et al., 2019), at both the conceptual and implementation levels. Here, we distinguish DKs from Deformable Convolutions and show how they can be unified.

Deformable Convolutions can be reformulated in a general form as

$$O_j = \sum_{k \in \mathcal{K}} I_{j+k+\Delta j} W_k, \tag{9}$$

where they aim to learn a group of data offsets $\{\Delta j\}$ with respect to discrete data positions $j$. For consistency for analysis, the value of effective receptive field becomes

$$\mathcal{R}_{\text{DC}}^{(n)}(i; j) = \sum_{k_m \in \mathcal{K}} \mathcal{R}^{(n)}(i; j + k_m + \Delta j_m, k_m). \tag{10}$$

This approach essentially recomposes the input image towards common modes such that semantic recognition suffers less from deformation. Moreover, according to our previous analysis in Equation 5, sampling data is another way of sampling the ERF. This, to a certain extent, also explains why Deformable Convolutions are well suited for learning deformation-agnostic representations.

Moreover, we can learn both data *and* kernel offsets in one convolutional operator. Conceptually, this can be done by merging Equation 7 with Equation 9, which leads to

$$O_j = \sum_{k \in \mathcal{K}} I_{j+k+\Delta j} W_{k+\Delta k},$$
$$\mathcal{R}_{\text{DC+DK}}^{(n)}(i; j) = \sum_{k_m \in \mathcal{K}} \mathcal{R}^{(n)}(i; j + k_m + \Delta j_m, k_m + \Delta k_m). \tag{11}$$

We also investigate this operator in our experiments. Although the two techniques may be viewed as serving a similar purpose, we find the collaboration between Deformable Kernels and Deformable Convolutions to be powerful in practice, suggesting strong compatibility.

## 4 EXPERIMENTS

We evaluate our Deformable Kernels (DKs) on image classification using ILSVRC and object detection using the COCO benchmark. Necessary details are provided to reproduce our results, together with descriptions on base models and strong baselines for all experiments and ablations. For task-specific considerations, we refer to each corresponding section.

**Implementation Details:** We implement our operators in PyTorch and CUDA. We exploit depth-wise convolutions when designing our operator for better computational efficiency[3]. We initialize kernel grids to be uniformly distributed within the scope size. For the kernel offset generator, we set its learning rate to be a fraction of that of the main network, which we cross-validate for each base model. We also find it important to clip sampling locations inside the original kernel space, such that $k + \Delta k \in \mathcal{K}$ in Equation 7.

**Base Models:** We choose our base models to be ResNet-50 (He et al., 2016) and MobileNet-V2 (Sandler et al., 2018), following the standard practice for most vision applications. As mentioned, we exploit depthwise convolution and thus make changes to the ResNet model. Concretely, we define our *ResNet-50-DW* base model by replacing all $3 \times 3$ convolutions by its depthwise counterpart while doubling the dimension of intermediate channels in all residual blocks. We find it to be a reasonable base model compared to the original ResNet-50, with comparable performance on both tasks. During training, we set the weight decay to be $4 \times 10^{-5}$ rather than the common $10^{-4}$ for both models since depthwise models usually underfit rather than overfit (Xie et al., 2017; Howard et al., 2017; Hu et al., 2018). We set the learning rate multiplier of DK operators as $10^{-2}$ for ResNet-50-DW and $10^{-1}$ for MobileNet-V2 in all of our experiments.

**Strong Baselines:** We develop our comparison with two previous works: Conditional Convolutions (Yang et al., 2019) for dynamics inference, and Deformable Convolutions (Dai et al., 2017; Zhu et al., 2019) for deformation modeling. We choose Conditional Convolutions due to similar computation forms – sampling can be deemed as an elementewise "expert voting" mechanism. For fair comparisons, We reimplement and reproduce their results. We also combine our operator with these previous approach to show both quantitative evidence and qualitative insight that our working mechanisms are compatible.

### 4.1 IMAGE CLASSIFICATION

We first train our networks on the ImageNet 2012 training set (Deng et al., 2009). We adopt a common experiment protocol for fair comparisons as in Goyal et al. (2017); Loshchilov & Hutter (2017). For more details, please refer to our supplement.

We first ablate the scope size of kernels for our DKs and study how it can affect model performance using ResNet-50-DW. As shown in Table 1, our DKs are sensitive to the choice of the scope size. We shown that when only applied to $3 \times 3$ convolutions inside residual bottlenecks, local DKs induce a +0.7 performance gain within the original scope. By further enlarging the scope size, performance increases yet quickly plateaus at scope $4 \times 4$, yielding largest +1.4 gain for top-1 accuracy. Our speculation is that, although increasing scope size theoretically means better interpolation, it also makes the optimization space exponentially larger for each convolutional layer. And since number of entries for updating is fixed, this also leads to relatively sparse gradient flows. In principle, we set default scope size at $4 \times 4$ for our DKs.

We next move on and ablate our designs by comparing the global DK with the local DK, shown in the table. Both operators helps while the local variants consistently performs better than their global counterparts, bringing a +0.5 gap on both base models. We also study the effect of using more DKs in the models – the $1 \times 1$ convolutions are replaced by global DKs[4] with scope $2 \times 2$. Note that all $1 \times 1$ convolutions are not depthwise, and therefore this operation induces nearly 4 times of parameters. We refer their results only for ablation and show that adding more DKs still helps – especially for MobileNet-V2 since it is under-parameterized. This finding also holds for previous models (Yang et al., 2019) as well.

---

[3] This makes enlarging the kernel scope size tractable and prevents extensive resource competition in CUDA kernels when applying local DKs.

[4] The implementation of local DKs right now cannot support large number of output channels.

| Backbone | 1×1 Deformable Kernels | 3×3 Deformable Kernels | top1 (%) | #P (M) | GFLOPs |
|---|---|---|---|---|---|
| ResNet-50 | w/o | w/o | 76.7 | 25.6 | 3.86 |
| ResNet-50-DW | w/o | w/o | 76.3 | 23.7 | 3.82 |
| ResNet-50-DW | w/o | local, scope size 3×3 | 77.4 | 24.9 | 4.32 |
| | | **local, scope size 4×4** | **78.1** | **25.0** | **4.32** |
| | | local, scope size 5×5 | 77.8 | 25.0 | 4.32 |
| | | local, scope size 9×9 | 77.4 | 25.4 | 4.32 |
| ResNet-50-DW | w/o | global, scope size 4×4 | 77.6 | 23.9 | 3.82 |
| | global, scope size 2×2 | global, scope size 4×4 | 77.9 | 80.1 | 4.09 |
| | w/o | local, scope size 4×4 | 78.1 | 25.0 | 4.32 |
| | **global, scope size 2×2** | **local, scope size 4×4** | **78.5** | **81.2** | **4.60** |
| MobileNet-V2 | w/o | w/o | 71.9 | 3.5 | 0.31 |
| MobileNet-V2 | w/o | global, scope size 4×4 | 73.6 | 3.7 | 0.31 |
| | global, scope size 2×2 | global, scope size 4×4 | 74.5 | 10.1 | 0.34 |
| | w/o | local, scope size 4×4 | 74.1 | 4.7 | 0.73 |
| | **global, scope size 2×2** | **local, scope size 4×4** | **74.8** | **11.1** | **0.76** |

Table 1: **Ablations of scope size and different instantiations of DK for image classification.** Using proper scope size, and more DK layers boosts performance. Modeling individual offset kernel grid for each data entries is also beneficial.

| Backbone | 1×1 Deformable Kernels | 3×3 Deformable Kernels | top1 (%) | #P (M) | GFLOPs |
|---|---|---|---|---|---|
| ResNet-50-DW | w/o | local, scope size 4×4 | 78.1 | 25.0 | 4.32 |
| ResNet-50-DW with SCC | w/o | w/o | 77.6 | 42.5 | 7.13 |
| | | local, scope size 4×4 | 78.9 | 43.7 | 7.61 |
| ResNet-50-DW with DCN | w/o | w/o | 78.0 | 24.8 | 4.10 |
| | | **local, scope size 4×4** | **79.0** | **26.1** | **4.60** |
| MobileNet-V2 | w/o | local, scope size 4×4 | 74.1 | 4.7 | 0.73 |
| MobileNet-V2 with SCC | w/o | w/o | 74.3 | 19.0 | 2.19 |
| | | **local, scope size 4×4** | **75.5** | **19.7** | **2.48** |
| MobileNet-V2 with DCN | w/o | w/o | 73.2 | 4.6 | 0.52 |
| | | local, scope size 4×4 | 74.4 | 5.8 | 0.93 |

Table 2: **Comparisons to strong baselines for image classification** DKs perform comparably or superiorly to previous methods. Further combinations yield consistent gain, suggesting orthogonal and compatible working mechanisms.

We further compare and combine DKs with Conditional Convolutions and Deformable Convolutions. Results are recorded in Table 2. We can see that DKs perform comparably on ResNet-V2 and compare favorably on MobileNet-V2 – improve +0.9 from Deformable Convolutions and achieve comparable results with less than a quarter number of parameters compared to Conditional Convolutions. Remarkably, we also show that if combined together, even larger performance gains are in reach. We see consistent boost in top-1 accuracy compared to strong baselines: +1.3/+1.0 on ResNet-50-DW, and +1.2/+1.2 on MobileNet-V2. These gaps are bigger than those from our own ablation, suggesting the working mechanisms across the operators to be orthogonal and compatible.

## 4.2 OBJECT DETECTION

We examine DKs on the COCO benchmark (Lin et al., 2014). For all experiments, we use Faster R-CNN (Ren et al., 2015) with FPN (Lin et al., 2017) as the base detector, plugging in the backbones we previously trained on ImageNet. For MobileNet-V2, we last feature maps of the each resolution for FPN post aggregation. Following the standard protocol, training and evaluation are performed on the 120k images in the `train-val` split and the 20k images in the `test-dev` split, respectively. For evaluation, we measure the standard mean average precision (mAP) and shattered scores for small, medium and large objects.

Table 3 and Table 4 follow the same style of analysis as in image classification. While the baseline methods of ResNet achieve 36.6 mAP, indicating a strong baseline detector, applying local DKs brings a +1.2 mAP improvement when replacing 3x3 rigid kernels alone and a +1.8 mAP improve-

| Backbone | 1×1 Deformable Kernels | 3×3 Deformable Kernels | mAP | mAP$_S$ | mAP$_M$ | mAP$_L$ |
|---|---|---|---|---|---|---|
| ResNet-50-DW | w/o | w/o | 36.6 | 22.1 | 39.9 | 46.6 |
| ResNet-50-DW | w/o | global, scope size 4×4 | 36.7 | 22.6 | 40.2 | 46.9 |
| | global, scope size 2×2 | global, scope size 4×4 | 37.1 | 23.1 | 40.6 | 46.6 |
| | w/o | local, scope size 4×4 | 37.8 | 23.4 | 41.6 | 48.2 |
| | **global, scope size 2×2** | **local, scope size 4×4** | **38.4** | **23.4** | **42.0** | **49.4** |
| MobileNet-V2 | w/o | w/o | 31.3 | 18.6 | 33.7 | 40.4 |
| MobileNet-V2 | w/o | global, scope size 4×4 | 32.5 | 19.6 | 35.8 | 41.9 |
| | global, scope size 2×2 | global, scope size 4×4 | 32.9 | 19.4 | 35.5 | 42.8 |
| | w/o | local, scope size 4×4 | 32.9 | 19.5 | 36.0 | 42.5 |
| | **global, scope size 2×2** | **local, scope size 4×4** | **33.7** | **20.2** | **36.7** | **44.0** |

Table 3: **Ablations for object detection.** Consistent results with image classification.

| Backbone | 1×1 Deformable Kernels | 3×3 Deformable Kernels | mAP | mAP$_S$ | mAP$_M$ | mAP$_L$ |
|---|---|---|---|---|---|---|
| ResNet-50-DW | global, scope size 2×2 | local, scope size 4×4 | 38.4 | 23.4 | 42.0 | 49.4 |
| ResNet-50-DW with SCC | w/o | w/o | 36.3 | 22.1 | 39.3 | 47.0 |
| | | local, scope size 4×4 | 38.0 | 23.4 | 41.9 | 48.4 |
| ResNet-50-DW with DCN | w/o | w/o | 39.9 | 24.0 | 43.4 | 52.6 |
| | | **local, scope size 4×4** | **40.6** | **24.6** | **43.9** | **53.3** |
| MobileNet-V2 | global, scope size 2×2 | local, scope size 4×4 | 33.7 | 20.2 | 36.7 | 44.0 |
| MobileNet-V2 with SCC | w/o | w/o | 33.2 | 20.5 | 35.6 | 43.3 |
| | | local, scope size 4×4 | 34.3 | 20.2 | 37.3 | 44.7 |
| MobileNet-V2 with DCN | w/o | w/o | 34.4 | 20.5 | 37.0 | 44.7 |
| | | **local, scope size 4×4** | **35.6** | **20.6** | **38.5** | **47.3** |

Table 4: **Comparisons to strong baselines for object detection** DKs perform fall short to Deformable Convolution, but combination still improves performance.

ment when replacing both 1x1 and 3x3 rigid kernels. This trend magnifies on MobileNet-v2 models, where we see an improvement of +1.6 mAP and +2.4 mAP, respectively. Results also confirm the effectiveness of local DKs against global DKs, which is again in line with our expectation that local DKs can model locality better.

For the comparisons with strong baselines, an interesting phenomenon worth noting is that though DKs perform better than Deformable Convolutions on image classification, they fall noticeably short for object detection measured by mAP. We speculate that even though both techniques can adapt ERF in theory (as justified in Section 3.2), directly shifting sampling locations on data is easier to optimize. Yet after combining DKs with previous approaches, we can consistently boost performance for all the methods – +0.7/+1.2 for Deformable Convolutions on each base models, and +1.7/+1.1 for Conditional Convolutions. These findings align with the results from image classification. We next investigate what DKs learn and why they are compatible with previous methods in general.

### 4.3    WHAT DO DEFORMABLE KERNELS LEARN?

**Awareness of Object Scale:** Since deformation is hard to quantify, we use object scale as a rough proxy to understand what DKs learn. In Figure 3, we show the t-SNE (Maaten & Hinton, 2008) of learned model dynamics by the last convolutional layers in MobileNet-V2 using Conditional Convolution and our DKs. We validate the finding as claimed by Yang et al. (2019) that the experts of Conditional Convolutions have better correlation with object semantics than their scales (in reference to Figure 6 from their paper). Instead, our DKs learn kernel sampling offsets that strongly correlate to scales rather than semantics. This sheds light on why the two operators are complementary in our previous experiments.

**Adaptation of Effective Receptive Fields:** To verify our claim that DK indeed adapts ERFs in practice, we show ERF visualizations on a set of images in which they display different degrees of deformations. We compare the results of rigid kernels, Deformable Convolutions, our DKs, and the combination of the two operators. For all examples, note that the theoretical receptive field covers every pixel in the image but ERFs contain only a central portion of it. Deformable Convolutions

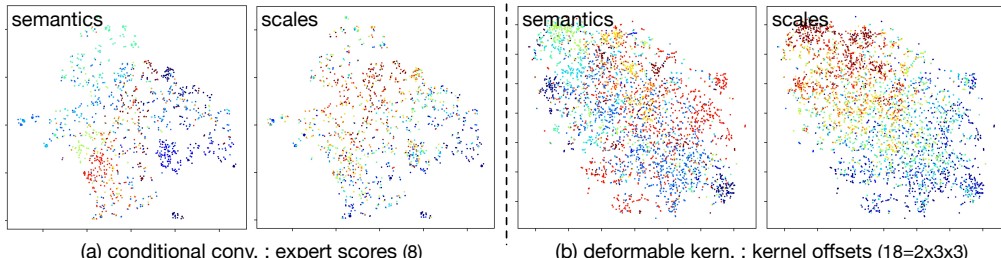

Figure 3: **Semantics vs. Scales.** We show t-SNE results of learned model dynamics using 10 random classes of objects from the COCO `test-dev` split. Each point represents an object extracted by ground-truth bounding box, whose color either denotes its class label or bounding box scale. The color of an object scale is its normalized area rank discretized by every 10th percentile among all data. Numbers inside parentheses indicate the dimension of learned dynamics before t-SNE. **(a)** The dynamics of Conditional Convolutions are closer to semantics than to object scales. **(b)** On the contrary, our DKs learn dynamics that are significantly related to scales rather than semantics.

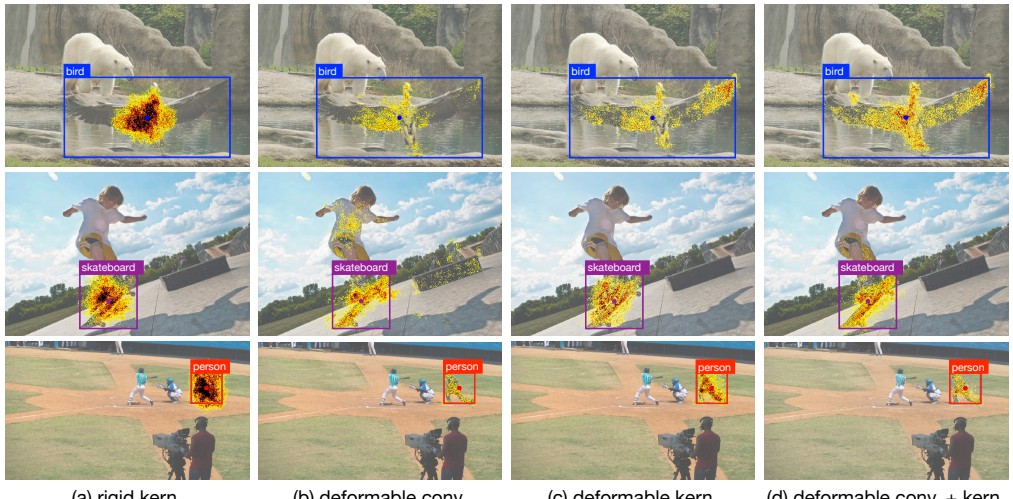

Figure 4: **Learned Effective Receptive Fields.** We show learned ERFs on three images with large, medium, and small objects from the COCO `test-dev` split. Given each ground-truth bounding box, we visualize the non-zero ERF values of its central point. Theoretical RFs cover the whole image for all three examples and we thus ignore them in our plots. **(a)** Rigid kernels have strong central effects and a Gaussian-like ERF that cannot deal with object deformation alone. **(b)** Deformable Convolutions and **(c)** Deformable Kernels both tune ERFs to data. **(d)** Combining both operators together enables better modeling of 2D geometric transformation of objects.

and DKs perform similarly in terms of adapting ERFs, but Deformable Convolutions tend to spread out and have sparse responses while DKs tend to concentrate and densely activate within an object region. Combining both operators yields more consistent ERFs that exploit both of their merits.

## 5 CONCLUSION

In this paper, we introduced Deformable Kernels (DKs) to adapt effective receptive fields (ERFs) of convolutional networks for object deformation. We proposed to sample kernel values from the original kernel space. This in effect samples the ERF in linear networks and also roughly generalizes to non-linear cases. We instantiated two variants of DKs and validate our designs, showing connections to previous works. Consistent improvements over them and compatibility with them were found, as illustrated in visualizations.

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

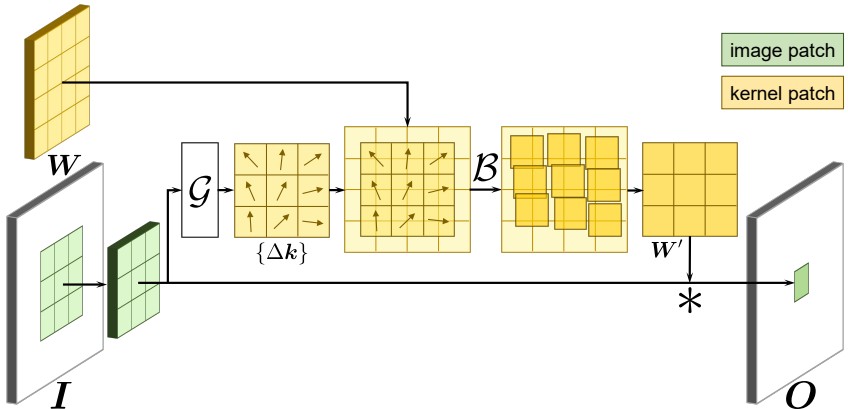

Figure 5: **Illustration of feed-forwarding through a 3×3 local Deformable Kernel from a 4×4 scope.** For each input patch, local DK first generates a group of kernel offsets $\{\Delta k\}$ from input feature patch using the light-weight generator $\mathcal{G}$ (a 3×3 convolution of rigid kernel). Given the original kernel weights $W$ and the offset group $\{\Delta k\}$, DK samples a new set of kernel $W'$ using a bilinear sampler $\mathcal{B}$. Finally, DK convolves the input feature map and the sampled kernels to complete the whole computation.

## A   COMPUTATION FLOW OF DEFORMABLE KERNELS

We now cover more details on implementing DKs by elaborating the computation flow of their forward and backward passes. We will focus on the local DK given its superior performance in practice. The extension to global DK implementation is straight-forward.

### A.1   FORWARD PASS

In Section 3.3, we introduce a kernel offset generator $\mathcal{G}$ and a bilinear sampler $\mathcal{B}$. Figure 5 illustrates an example of the forward pass.

Concretely, given a kernel $W$ and a learned group of kernel offsets $\{\Delta k\}$ on top of a regular 2D grid $\{k\}$, we can resample a new kernel $W'$ by a bilinear operator $\mathcal{B}$ as

$$W' \equiv W_{k+\Delta k} = \sum_{k' \in \mathcal{K}} \mathcal{B}(k + \Delta k, k') W_{k'}, \tag{12}$$

$$\text{where } \mathcal{B}(k + \Delta k, k') = \max(0, 1 - |k_x + \Delta k_x - k'_x|) \cdot \max(0, 1 - |k_y + \Delta k_y - k'_y|).$$

Given this resampled kernel, DK convolves it with the input image just as in normal convolutions using rigid kernels, characterized by Equation 1.

### A.2   BACKWARD PASS

The backward pass of local DK consists of three types of gradients: (1) the gradient to the data of the previous layer, (2) the gradient to the full scope kernel of the current layer and (3) the additional gradient to the kernel offset generator of the current layer. The first two types of gradients share same forms of the computation comparing to the normal convolutions. We now cover the computation for the third flow of gradient that directs where to sample kernel values.

In the context of Equation 7, the partial derivative of a output item $O_j$ w.r.t. $x$ component of a given kernel offset $\Delta k_x$ (similar for its $y$ component $\Delta k_y$) can be computed as

$$\frac{\partial O_j}{\partial \Delta k_x} = \sum_k I_{j+k} \left( \sum_{k'} W_{k'} \frac{\partial \mathcal{B}(k + \Delta k, k')}{\partial \Delta k_x} \right), \tag{13}$$

$$\text{where } \frac{\partial \mathcal{B}(k + \Delta k, k')}{\partial \Delta k_x} = \max(0, 1 - |k_y + \Delta k_y - k'_y|) \cdot \begin{cases} 0 & |k_x + \Delta k_x - k'_x| \geq 1 \\ 1 & k_x + \Delta k_x < k'_x \\ -1 & k_x + \Delta k_x \geq k'_x \end{cases}.$$

| Output | ResNet-50 | | ResNet-50-DW | |
|---|---|---|---|---|
| $112 \times 112$ | $7 \times 7$, 64, stride 2 | | | |
| $56 \times 56$ | $3 \times 3$ max pool, stride 2 | | | |
| $56 \times 56$ | $\begin{array}{l} 1 \times 1, 64 \\ 3 \times 3, 64 \\ 1 \times 1, 256 \end{array}$ | $\times 3$ | $\begin{array}{l} 1 \times 1, 128 \\ 3 \times 3, 128, G = 128 \\ 1 \times 1, 256 \end{array}$ | $\times 3$ |
| $28 \times 28$ | $\begin{array}{l} 1 \times 1, 128 \\ 3 \times 3, 128 \\ 1 \times 1, 512 \end{array}$ | $\times 4$ | $\begin{array}{l} 1 \times 1, 256 \\ 3 \times 3, 256, G = 256 \\ 1 \times 1, 512 \end{array}$ | $\times 4$ |
| $14 \times 14$ | $\begin{array}{l} 1 \times 1, 256 \\ 3 \times 3, 256 \\ 1 \times 1, 1024 \end{array}$ | $\times 6$ | $\begin{array}{l} 1 \times 1, 512 \\ 3 \times 3, 512, G = 512 \\ 1 \times 1, 1024 \end{array}$ | $\times 6$ |
| $7 \times 7$ | $\begin{array}{l} 1 \times 1, 512 \\ 3 \times 3, 512 \\ 1 \times 1, 2048 \end{array}$ | $\times 3$ | $\begin{array}{l} 1 \times 1, 1024 \\ 3 \times 3, 1024, G = 1024 \\ 1 \times 1, 2048 \end{array}$ | $\times 3$ |
| $1 \times 1$ | $7 \times 7$ global average pool, 1000-d $fc$, softmax | | | |
| #P (M) | 25.6 | | 23.7 | |
| GFLOPs | 3.86 | | 3.82 | |

Table 5: **Network architecture of our ResNet-50-DW comparing to the original ResNet-50** Inside the brackets are the general shape of a residual block, including filter sizes and feature dimensionalities. The number of stacked blocks on each stage is presented outside the brackets. "$G = 128$" suggests the depthwise convolution with 128 input channels. Two models have similar numbers of parameters and FLOPs. At the same time, depthwise convolutions facilitate the computation efficiency of our Deformable Kernels.

## B  NETWORK ARCHITECTURES

Table 5 shows the comparison between the original ResNet-50 (He et al., 2016) and our modified ResNet-50-DW. The motivation of introducing depthwise convolutions to ResNet is to accelerate the computation of local DKs based on our current implementations. The ResNet-50-DW model has similar model capacity/complexity and performance (see Table 1) compared to its non-depthwise counterpart, making it an ideal base architecture for our experiments.

On the other hand, in all of our experiments, MobileNet-V2 (Sandler et al., 2018) base model is left untouched.

## C  ADDITIONAL COMPARISON OF EFFECTIVE RECEPTIVE FIELDS

We here show additional comparison of ERFs when objects have different kinds of deformations in Figure 6. Comparing to baseline, our method can adapt ERFs to be more persistent to object's semantic rather than its geometric configuration.

## D  ADDITIONAL EXPERIMENT DETAILS

**Image Classification:** Similar to Goyal et al. (2017); Loshchilov & Hutter (2017), training is performed by SGD for 90 epochs with momentum 0.9 and batch size 256. We set our learning rate of $10^{-1}$ so that it linearly warms up from zero within first 5 epochs. A cosine training schedule is applied over the training epochs. We use scale and aspect ratio augmentation with color perturbation as standard data augmentations. We evaluate the performance of trained models on the ImageNet 2012 validation set. The images are resized so that the shorter side is of 256 pixels. We then centrally crop $224 \times 224$ windows from the images as input to measure recognition accuracy.

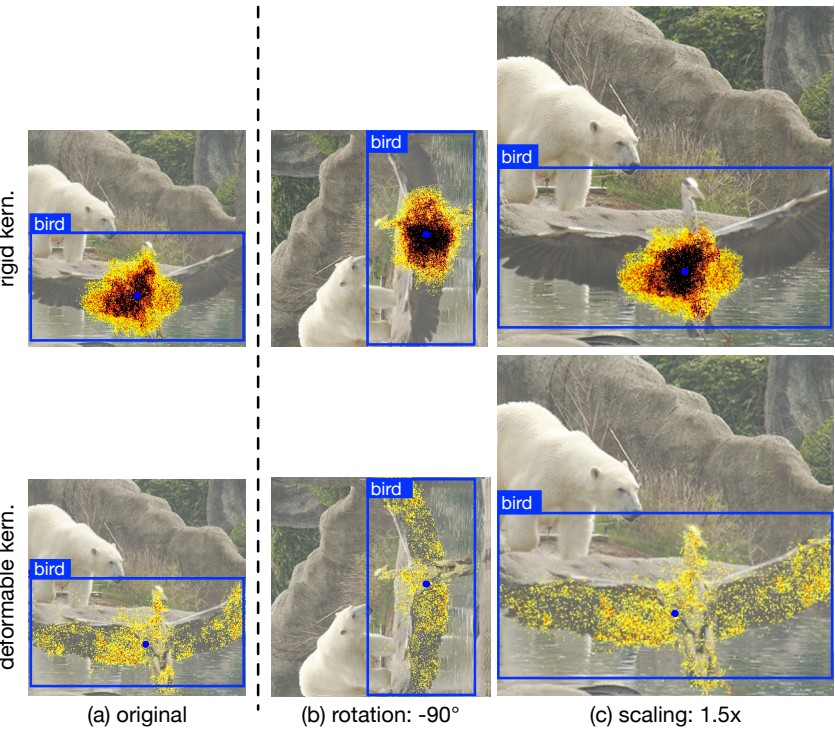

Figure 6: **Effective Receptive Field Comparison between rigid kernels and DKs under different kinds of Object Deformation.** At each row and from left to right, we show the original image (1300×800), the image rotated by -90 degrees and the image scaled by 1.5 times. Images are cropped and resized for the typesetting purpose.

