# OpenReview forum: "Deformable Kernels: Adapting Effective Receptive Fields for Object Deformation"
_ICLR.cc/2020/Conference — Accept (Poster)_

### Official Review · AnonReviewer3 · 2019-10-23
**Official Blind Review #3**

**Rating:** 6

**Review:**

Updates:
Thanks for the updates and I appreciate the authors' effort in running new experiments, whose results are very interesting and inspiring to me.

My score remains unchanged. The main reason is that I am not an expert in this subject.


--------------------------------------------
Traditional convolution neural networks are not aware of object’s geometric variations (i.e. rotation), while human being are very good at abstracting out such variation.
In this paper, the authors propose an approach known as DKs (deformation kernels) to overcome such issues. The high level picture is make the convolutional filter data dependable. For convnets, the filter is independent of the data. To make it data dependent (potentially able to detect the objects geometric information), DKs first initialize a larger filter (say 9 * 9) that is universal to all inputs and then `subsampling` a smaller filter (say 3 *3), the `subsampling` strategy is input dependent and also learnable (similar to attention mechanism.)

I do not have much background in this field, but I found the ideas of DKs very interesting and novel (assuming this is the first work to make the filter data dependent and learnable.)  I lean to a weakly accept.

Minor Comments:
1. Below equation (1). Z  -> Z^2; above (1) : j \in R^2  -> Z^2
2. above (5) as a composition --> as a sum ??
3. can you elaborate on 'In practice, we use the bilinear sampling operator to interpolate within the discrete kernel grid.' In particular, how the `smalle`r kernel is learned/subsampled from the larger kernel? (i.e. how \Delta k is learned?)

More interesting experiments?
1. Compare performance of the two methods below:
 a. standard architectures using data deformation (translation, rotation, dilation)
 b. DKs without applying data deformation.
2. Figure 4.  Could you produce plots similar to the setting of figure 1., i.e. rotate / dilate the images. Showing that DKs could effectively capture such deformation.



**Experience Assessment:**

I do not know much about this area.

**Review Assessment: Checking Correctness Of Derivations And Theory:**

N/A

**Review Assessment: Checking Correctness Of Experiments:**

N/A

**Review Assessment: Thoroughness In Paper Reading:**

N/A

---

> ### Author Response · Authors · 2019-11-14
> **Response to Reviewer 3**
>
> Thanks for your review and suggestion to improve our work. Before we step forward to answer your questions, we want to first clarify on your comment “assuming this is the first work to make the filter data-dependent and learnable“. This is unfortunately not true. As we mentioned in Section 2 (or more specifically, the “Dynamic Inference“ subsection), there are works that “adapt convolutional kernels at runtime“ and “can learn and infer customized kernel spaces with respect to the data“. Our DKs differ from these works with the emphasis on directly adapting ERFs by spatial subsampling. We also compared with one method from such line of works, i.e., Conditional Convolutions [Yang et al., 2019], which is referred to as SCC in our experiments. Figure 3 shows that the dynamics of SCC are closer to semantics, while our DKs learn dynamics that are significantly related to scales. We will underscore the difference between our DKs and previous works from Dynamic Inference in our next revision.
>
> ----------
> Q1. “Above (5) as a composition --> as a sum?? ”
> A1. The term “composition“ may be ambiguous here and lead to confusion. In fact, a KxK kernel can be deemed as a sum of $K^2$ shifted 1x1 kernels according to its kernel position. We will make this point clearer and rewrite this specific sentence in our next version.
>
> ----------
> Q2. “How is the 'smaller' kernel learned/subsampled from the larger kernel? (i.e., how is $\Delta k$ learned?)”
> A2. In a nutshell, we follow a very similar computational flow as in Deformable Convolutions [Dai et al., 2017; Zhu et al., 2019]. We have added a new section in Appendix A to resolve your concern -- hopefully, it could help make things clearer. Meanwhile, R2 has similar concerns regarding our implementation of DK. Please refer to our response to R2-A2 for more details as well.
>
> ----------
> Q3. “More interesting experiments?”
>
> Q3(1). “Standard architectures using data deformation (translation, rotation, dilation)”
> A3(1). Thanks for your suggestion. And actually, our baselines on ImageNet (also as in the standard ImageNet training protocol) already use certain kinds of data deformation, such as random cropped resizing that simulate scaling and shifting deformation.
>
> Q3(2). “DKs without applying data deformation.”
> A3(2). Thanks for your suggestion. Yet we want to argue that the experiments without applying data augmentation actually cannot provide much insight. Since training on ImageNet with scaling data augmentation is crucial for high accuracy, the result of DKs without such augmentation is predictable to be much worse. Yet this does not necessarily mean that our DKs cannot learn deformation modalities. On the other hand, also because of the optimization problem we mentioned in R2-A3, our DKs still need a lot of data capacity to learn good adaptation of object deformations. We deem our DKs as a probe to enable convolutions to explicitly model deformation. However, to make them work nicely in practice, practitioners need good training strategies, and applying data deformation is one of them.
>
> Q3(3). “Could you produce plots similar to the setting of Figure 1.”
> A3(3). We add a new plot in Figure 6 (Appendix C) to compare rigid kernels and our DKs under rotation and scaling conditions, respectively. Our DKs can learn persistent feature encoding of images that is not sensitive to data deformation.

---

### Official Review · AnonReviewer2 · 2019-10-25
**Official Blind Review #2**

**Rating:** 6

**Review:**

This work presents the idea of deformable kernels (DKs). As opposed to rigid kernels in standard convolutional networks, DKs allow each of their grid locations to be moved around in a larger kernel field. The offset by which a DK grid cell is moved is computed conditioned on the input to the network. To motivate the idea of DKs, the authors give some background on convolution, receptive and effective receptive fields (ERFs). The authors argue that since ERFs are spatially porous and irregularly distributed, one way to model them is to convolve square grids of input with DKs, which are composed of samples drawn from larger kernels. The authors define the concept of global and local DKs. They further contrast DKs with spatial sampling (deformable convolutions) and argue that although conceptually similar, both approaches are complementary to each other and can be used in combination in practice. Numerical experiments show competitive performance of DKs on image classification and object detection tasks. In the end empirical analysis is performed to analyze the characteristics of DKs.

I am unfamiliar with prior work in this direction, but the idea of DKs seems to be conceptually appealing and as the authors point-out, their approach can be seen as an alternative to spatial sampling for modeling deformations. Unfortunately the authors get hand-wavy when in Sec 3.2, they claim that the idea of subsampling kernels "roughly generalizes" to non-linear networks. I don't see how they can generalize what they present beyond piece-wise linear networks. I appreciate the effort to give a background on ERFs and describe (local and global) DKs, but in my opinion, technical sections of the paper partly very obscure. For instance it is not entirely clear how the offset predictors are trained, how exactly the sampling is used, details of architecture etc.

Empirically the method does not seem to offer a significant performance boost. Also, while the authors sell the idea of subsampling kernels, but the finding that kernel sizes beyond 4x4 don't seem to offer any benefit make the idea practically questionable.

The idea of DKs seems relevant, but both conceptually and empirically it seems very close to deformable convolutions. The authors need to clearly present their work, including its shortcomings (i.e., generalization or not beyond linear networks).

**Experience Assessment:**

I do not know much about this area.

**Review Assessment: Checking Correctness Of Derivations And Theory:**

I assessed the sensibility of the derivations and theory.

**Review Assessment: Checking Correctness Of Experiments:**

I assessed the sensibility of the experiments.

**Review Assessment: Thoroughness In Paper Reading:**

I read the paper thoroughly.

---

> ### Author Response · Authors · 2019-11-14
> **Response to Reviewer 2**
>
> We would like to thank you for your constructive comments, and please see below our response to each of your concerns.
>
> ----------
> Q1. “Unfortunately the authors get hand-wavy when in Sec 3.2, they claim that the idea of subsampling kernels 'roughly generalizes' to non-linear networks. I don't see how they can generalize what they present beyond piece-wise linear networks.”
> A1. To our understanding, ReLU networks are the piece-wise linear networks that you may be referring to. If so, that is indeed the case -- our current analysis can only generalize to ReLU networks, but not to other non-linear networks. However, it should be noted that since most of the currently prevalent models are actual ReLU networks, our analysis is broadly applicable. On the other hand, for other non-linear networks, the $C^{(n)}$ term in the equation at the end of Page 4 would become a cumulative product of corresponding non-linear activation derivatives, which is not simply a binary switch and thus hard to analysis. We will revise this part as well as make our point clearer in the next version. Thanks for your suggestion.
>
> ----------
> Q2. “It is not entirely clear how the offset predictors are trained, how exactly the sampling is used, details of architecture, etc.”
> A2. We follow Deformable Convolution (DC) [Dai et al., 2017; Zhu et al., 2019] on computing kernel offsets. For each input feature map of Deformable Kernel (DK), we use a 3x3 convolution with 18 dimensional output (3x3 points with x and y coordinate offsets) to generate the kernel offsets. Since our offsets are not integers, we use a differentiable bilinear sampling to sample kernels from non-integer coordinates (see Appendix A for detailed illustration on the forward and backward passes). On the other hand, our base network architectures are just modified ResNet (what we call ResNet-50-DW) and MobileNet-V2, respectively. To make this point clearer, we add more details in Table 5 of Appendix B. For all the experiments, we replace convolutions in each Residual Block with DKs, as discussed in Section 4.
>
> ----------
> Q3. “The finding that kernel sizes beyond 4x4 don't seem to offer any benefit makes the idea practically questionable.”
> A3. Indeed, intuitively, with a higher resolution of kernel space (what we called “scope size“), our bilinear sampler for generating kernel values should yield a better approximation of latent kernel values at given coordinates. Yet in practice, we observe the performance saturate at 4x4 scope size. We speculate that this issue may stem from the side of optimization: consider sampling a 3x3 kernel from a 9x9 kernel space; to learn a good 9x9 kernel space, it is reasonable to assume each specific kernel value shall consume a certain amount of data. Since our gradient is either sparse -- not all kernel values in the 9x9 space will get the gradient as opposed to rigid kernels, DKs will roughly require 9 = (9 x 9) / (3 x 3) times more data. In practice, this indicates that some of the kernels cannot be trained sufficiently, which can partly explain the phenomenon we have observed.

---

### Official Review · AnonReviewer1 · 2019-10-28
**Official Blind Review #1**

**Rating:** 6

**Review:**

This paper introduce a simple algorithm called deformable kernels. It learns to generate a collection of coordinate offset Δk for each of the convolutional kernel element. Then during convolution, the kernel is treated as a 2D regular grid and sampled (interpolated) according to the generated coordinate offset before applying to the inputs. An auxiliary shallow network is learned to generate those coordinate offsets based in inputs. This method is very similar to the existing "deformable convolution" algorithm, though this operate on the kernels instead. Numerical experiments on image classification and object detection tasks show that the method performs better or comparably to strong baselines. It boost the performance even more when combined with existing methods.

This paper is relatively easy to follow and the ideas are simple and effective.

My main concern about this paper is the novelty given its similarity to the previous methods. Maybe it could improve if more and in-depth studies are shown that analyze what deformable kernel learns and why that is different from what deformable convolution learn.

**Experience Assessment:**

I do not know much about this area.

**Review Assessment: Checking Correctness Of Derivations And Theory:**

N/A

**Review Assessment: Checking Correctness Of Experiments:**

I assessed the sensibility of the experiments.

**Review Assessment: Thoroughness In Paper Reading:**

N/A

---

> ### Author Response · Authors · 2019-11-14
> **Response to Reviewer 1**
>
>  Thanks for your review and comments. We here clarify to each of your inquiries accordingly:
>
> ----------
> Q1. “Novelty compared to previous methods.”
> A1. Our novelty is mainly threefold:
>     + the observation that, while most current deformation modeling approaches [Dai et al. 2017; Zhu et al., 2019], as well as some dynamic inference approaches [Li et al., 2019; Shelhamer et al., 2019], adapt the convolutional Receptive Field (RF) during runtime, the ultimate goal should actually be to manipulate the Effective Receptive Field (ERF) instead. This discrepancy motivates us to look closer at what factors control ERF.
>     + the insight and analysis of factors that *directly* control ERF — i.e., data sampling locations & kernel values — on top of and beyond the original proposal by Luo et al., 2016, which emphasizes expectation cases that cannot be leveraged to manipulate ERF arbitrarily (say 2x bigger). To the best of our knowledge, our paper is the first work that systematically discusses how we can actively manipulate ERF during runtime.
>     + a new family of convolutional operators, namely Deformable Kernel (DK), that leverages our theory to manipulate ERFs. In the range of possible designs that our theory suggests, we choose one for which it is possible to adapt the ERF without touching the RF in general -- strickly different from what Deformable Convolution (DC) [Dai et al. 2017; Zhu et al., 2019] does. We also prove with thorough experiments and qualitative insights that our DK can indeed help to model object deformations.
>
> To be more specific, we deem DC as our “nearest neighbor“ with respect to both general purpose and computational form. The main difference between DC and our DK is that, for each convolution, DC learns to sample the data space (see Equation 9, so as to change RF as well), while our DK learns to sample the kernel space (see Equation 7, so as to not change RF at all). Empirically we show that such a simple difference leads to different characteristics in practice. For example, visualization of ERFs learned by both operators (Figure 4b & 4c) shows that DCs encourage ERFs to spread out and have sparse responses while DKs tend to make them concentrated and densely activated within an object region.
>
> ----------
> Q2. “What does DK learn and why is it different from what DC learns?”
> A2. In Figure 3b, we show t-SNE results of learned dynamics (or kernel sampling offsets) from our local DK operators, and they are significantly related to scale -- one of the most common forms of object deformation in 2D space.
>
> In Figure 6 (Appendix C), we provide new experimental results to show that ERFs learned by DKs are nearly equivariant to rotation (-90°) and scaling (1.5x). Note that similar properties are also observed in DCs [Zhu et al., 2019] (see their Figure 5). So to answer your second question: what DK learns is very similar to what DC learns, even though computationally, they have different behaviors. And in fact, this is also our initial design intention, since being able to persistently fire to an object’s semantics rather than its possible geometric configuration is one of the most straightforward ways of dealing with deformation.

---

### Author Response · Authors · 2019-11-14
**Paper updated with more details in the Appendix**

We have updated our paper during the rebuttal session to include more details regarding to
    + implementation of the operators (Appendix A),
    + model specifics we used for experiments (Appendix B),
    + and more visualization of ERFs under different forms of object deformation (Appendix C).

We will make our code publicly available. Thank all for suggestions.

---

### Decision · Program_Chairs · 2019-12-19

**Decision:**

Accept (Poster)

**Comment:**

In my opinion, this paper is borderline (but my expertise is not in this area) and the reviewers are too uncertain to be of help in making an informed decision.